# Spatial Variation and Relation of Aerosol Optical Depth with LULC and Spectral Indices

Vipasha Sharma [1], Swagata Ghosh [1,*], Sultan Singh [2], Dinesh Kumar Vishwakarma [3,*], Nadhir Al-Ansari [4,*], Ravindra Kumar Tiwari [5] and Alban Kuriqi [6,7,*]

1 Amity Institute of Geoinformatics and Remote Sensing (AIGIRS), Amity University, Noida 201313, Uttar Pradesh, India
2 Haryana Space Applications Centre, Gurugram 122001, Haryana, India
3 Department of Irrigation and Drainage Engineering, G.B. Pant University of Agriculture and Technology, Pantnagar 263145, Uttarakhand, India
4 Civil, Environmental, and Natural Resources Engineering, Lulea University of Technology, 97187 Lulea, Sweden
5 Department of Food Technology and Nutrition, Lovely Professional University, Phagwara 144001, Punjab, India
6 CERIS, Instituto Superior T'ecnico, University of Lisbon, 1649-004 Lisbon, Portugal
7 Civil Engineering Department, University for Business and Technology, 10000 Pristina, Kosovo
* Correspondence: swagata.gis@gmail.com or sghosh1@amity.edu (S.G.); dinesh.vishwakarma4820@gmail.com (D.K.V.); nadhir.alansari@ltu.se (N.A.-A.); alban.kuriqi@tecnico.ulisboa.pt (A.K.)

**Abstract:** In the current study area (Faridabad, Gurugram, Ghaziabad, and Gautam Buddha Nagar), the aerosol concentration is very high, adversely affecting the environmental conditions and air quality. Investigating the impact of Land Use Land Cover (LULC) on Aerosol Optical Depth (AOD) helps us to develop effective solutions for improving air quality. Hence, the spectral indices derived from LULC ((Normalized difference vegetation index (NDVI), Soil adjusted vegetation index (SAVI), Enhanced vegetation index (EVI), and Normalized difference build-up index (NDBI)) with Moderate Resolution Imaging Spectroradiometer (MODIS) Multiangle Implementation of Atmospheric Correction (MAIAC) high spatial resolution (1 km) AOD from the years 2010–2019 (less to high urbanized period) has been correlated. The current study used remote sensing and Geographical Information System (GIS) techniques to examine changes in LULC in the current study region over the ten years (2010–2019) and the relationship between LULC and AOD. A significant increase in built-up areas (12.18%) and grasslands (51.29%) was observed during 2010–2019, while cropland decreased by 4.42%. A positive correlation between NDBI and SAVI (0.35, 0.27) indicates that built-up soils play an important role in accumulating AOD in a semi-arid region. At the same time, a negative correlation between NDVI and EVI ($-0.24$, $-0.15$) indicates the removal of aerosols due to an increase in vegetation. The results indicate that SAVI can play an important role in $PM_{2.5}$ modeling in semi-arid regions. Based on these findings, urban planners can improve land use management, air quality, and urban planning.

**Keywords:** MODIS; MAIAC; AOD; AERONET; LULC; NDVI; NCR

## 1. Introduction

Aerosols are multi-phased particles of both solid and liquid composition in the atmosphere. Natural processes or anthropogenic activities can lead to the formation of these pollutants. Aerosols, despite their small volume, can significantly alter the earth's environment and human life, e.g., aerosols affect the energy budget of the earth [1], the water cycle [2], monsoon patterns [3,4], crop yield and security of food [5,6], and reduces cloud cover [7], among others. The role of aerosols in climate change and atmospheric

radiation balance is also undeniable. Aerosols are important contributors to local, regional, and global climate change.

Moreover, aerosols induce the rate of mortality and morbidity [8–10]. Even an increase in aerosols might lead to many accidents on the road, as it also reduces visibility levels [11]. In urban areas, air quality parameters are often closely linked to urbanization [12]. Urbanization and economic growth leading to land use land cover (LULC) transformation mainly increase in the built-up area followed by fallow/open land and decrease in vegetation cover, agricultural land, and water bodies have resulted in increased emissions of air pollutants, resulting in a worsening of air quality, affecting regional climate and thereby influencing air pollution transport and diffusion [13–16]. The increased infrastructure leads to the built-up (high-rise buildings, roads, and highways, among others) increment and decreased cropland [17]. Looking into the wide-ranging impact of aerosols, monitoring the spatial and temporally aerosol concentration is essential.

Aerosol Optical Depth (AOD) measures the atmospheric aerosol and the degree of pollution in the air at a broad level. By measuring Aerosol Optical Depth (AOD), a Spatio-temporal assessment of aerosol concentration can be performed. The AERosol RObotic NETwork (AERONET) program provides periodic measurements of AOD, which develops insights about aerosol characteristics and spectral dependency, but with limited spatial coverage due to the dearth of functioning ground stations [18,19]. AOD acquisition through Remote Sensing (RS) offers unique advantages due to its ability to achieve high spatial and temporal resolutions over a wide geographic area. By using remote sensing, we can solve the gaps created by the absence or dispersion of weather observatories.

Furthermore, RS provides theoretical support for managing regional atmospheric environments due to its comprehensive understanding of aerosol concentrations and distributions. Based on Moderate Resolution Imaging Spectroradiometer (MODIS) data collection, AOD information is retrieved globally and provides daily or near-continuous time coverage [20–27]. Terra and Aqua are both integrated into the high-resolution MCD19A2 product, which uses the multi-angle implementation of the atmospheric correction (MAIAC) algorithm. AOD product obtained from the algorithm is characterized by a high resolution (1 km pixels), a wide range of inversion accuracy, and a wide range of inversion range [28–31]. The MAIAC algorithm has been preferred by current research on AOD retrieval because of the high retrieval accuracy and high resolution of MODIS AOD [20,32–34]. The potential of the MAIAC retrieved AOD in the region with limited in situ data has not been evaluated widely. Given this, it is necessary to verify the product and conduct a spatiotemporal analysis in the data-scarce region [19,35,36].

The present study area covers the constituent part of Delhi, the National Capital Region (NCR), which has undergone unprecedented economic developments and dramatic urbanization over the past decades [37]. There is a proposal for special economic zones (SEZ) in places like Noida and Gurugram. The expansion of industrial enterprises and the financial and private sectors are to blame for the increase in urbanization [13,17]. This has drawn a significant amount labor force to the region, eventually contributing to population expansion and changes in the LULC pattern of the current research area during 2010–2019. Rapid economic development triggers a noticeable change in LULC within a relatively short period in the study region. However, the research region includes the top 10 most polluted cities in the *World Air Quality Report* [38,39]. However, surprisingly few studies have concentrated on the mesoscale or local level of the current research region [40,41]. Therefore, the current study has thus attempted to provide a preliminary study in such a data-scarce zone. Various factors, including the meteorological environment and anthropogenic emissions, have contributed to the deterioration of air quality in the study area. In addition, LULC types may also have significant effects on surface properties and further affect regional meteorological conditions. To explore the impact of such changes on air quality, we systematically examined the correlation of several spectral indices representing the abundance of vegetation and built-up with AOD.

Detecting the geographical complexity of the distribution of AOD is possible through detailed information about the heterogeneity of a landscape composition by identifying the LULC. The AOD distribution and pattern are influenced by the LULC pattern, which is directly tied to the spatial distribution of the vegetation index and built-up index. The link between LULC and AOD is examined through the correlation between the AOD and spectral indices. NDVI, SAVI, EVI, and NDBI led to the quantitative association investigation. In the south and southeast Asian countries, several studies have examined AOD properties and their effects on global and local regions on a high spatial scale, mostly in China [42–47]. In the Indian scenario, only a handful of studies tested $MAIAC_{AOD}$, which were restricted mostly to the IGP and Delhi [32,48,49]. To the best of our knowledge, no studies explored the response of AOD to the LULC change in the Indian scenario.

Moreover, despite being critically polluted, $MAIAC_{AOD}$ has been rarely used in NCR aerosol data-scarce regions [48,49]. Nevertheless, the current study area is more active in creating and implementing regulations concerning LULC planning and air pollution management. Consequently, it is crucial to examine how LULC affects AOD at different points in the study area. Therefore, the present study attempts the validation of MAIAC AOD against AERONET AOD over selected constituent areas of the NCR and the correlation analysis of LULC and $AOD_{MAIAC}$.

As a general hypothesis, the land use land cover and the derived indices have been considered parameters for modeling $PM_{2.5}$. Due to the strong relationship between AOD and PM, the indices have been correlated with AOD prior to modeling to determine which parameter should be considered further. The present study seeks to determine the changes in LULC for the region, analyze the decadal variation of AOD for the region, validate satellite-based AOD using the AERONET AOD, and correlate AOD with LULC indices. The following correlation can help to identify the importance of parameters for modeling $PM_{2.5}$. Results from the study could be of use in air quality enhancement as a part of urban & rural planning and are expected to be beneficial in identifying micro-level pollution, aiding modeling communities, and algorithm developers in developing finer algorithms. This study can help to identify hot spots of polluted areas, which will help policymakers and real estate people to make a sustainable place for living. Furthermore, the results of this study will be useful for future urban planning and forecasting and controlling air pollution. The main objectives of the study are:

(1) To study the spatial variation of AOD in the current study area.
(2) To analyze the change in LULC from 2010 to 2019.
(3) To examine the correlation between AOD and LULC-derived indices.

## 2. Study Area

The present study area (28.07° N–28.92° N and 76.65° E–78.21° E) constitutes the districts of NCR, Faridabad, Ghaziabad, Gurugram, and Gautam Buddha Nagar (Figure 1). Presently, the rate of urbanization is 62.6% in the study area [50]. The current research identified three types of LULC classes: (1) Cropland, (2) Built-Up, and (3) Grassland. The present study area is dominated by heavy industrial pollution, vehicle emissions, fossil fuel burning, anthropogenic activities, and other factors due to rapid urbanization [51–55]. Gurugram, Ghaziabad, and Faridabad are among the top 10 polluted cities in the South Asian region [56]. Even though urbanization and air pollution are high in the current area, aerosol characteristics are still less explored. Pre-monsoon (March-May), monsoon (June–September), post-monsoon (October–November), and winter (December–February) are the four major seasons in India. It is a semi-arid region with tropical climates, hot summers (25 °C to 49 °C), and cold winters (22 °C to 2 °C) [57]. The in situ observations of AOD has been collected from the AERONET sites of Gurugram, Amity University (28.32° N, 76.92° E), and Gual Pahari (28.43° N, 77.15° E).

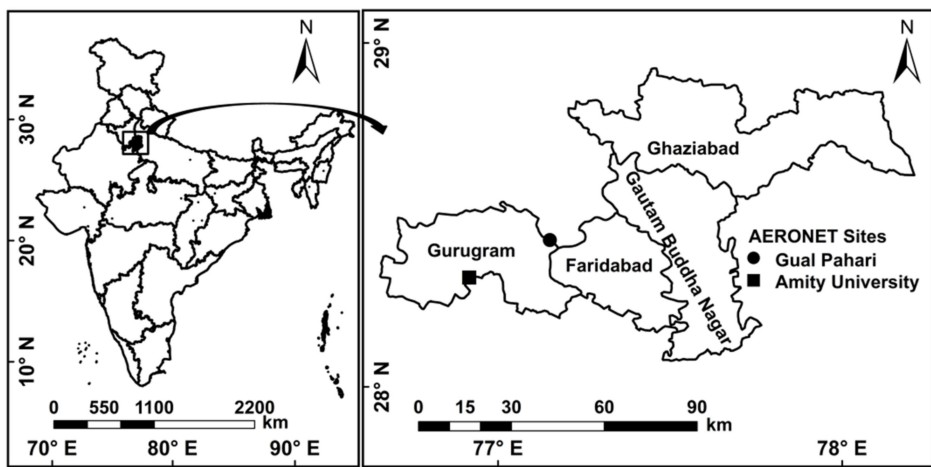

**Figure 1.** Map of the study area.

### 3. Data Used and Methodology

In the present research, AOD observations of MODIS ($AOD_{MAIAC}$) combined-Terra-Aqua collection 6 (C6) (MCD19A2) and in situ data ($AOD_{AERONET}$) from 2010 to 2019 have been utilized. The daily $AOD_{MAIAC}$ (550 nm) from 2010 to 2019 with 1 km spatial resolution has been used. In general, the current algorithm works as a combination of image processing and time series analysis. Details of the MAIAC algorithm are provided in the published literature [30,58]. AODs of the highest quality were utilized in the present study at 0.55 μm. The MAIAC AOD error envelope used in the algorithm was evaluated for accuracy using ± (0.05 + 15% * AOD) [20,30,59]. A current version of MAIAC is used in the present study, which is MCD19 with Collection 6 products.

The in-situ observations of AERONET are derived from a sunphotometer network that offers optical properties of aerosol worldwide at a fine resolution of AOD at 5–15 min intervals and a sky radiance at 30 min. The cloud screening of AOD and quality-control checked (Level 2.0) data were compared with $AOD_{MAIAC}$ for two sites of Gurugram, i.e., Gual Pahari and Amity University, from 2010 to 2019 with a data gap. The details of the data have been provided in Table 1. Figure 1 shows the location of the AERONET sites. $AOD_{MAIAC}$ is available at 550 nm and $AOD_{AERONET}$ at 500 nm, so to compare the data, AODAERONET interpolating to 550 nm was done using Angstrom's equation by using Angstrom Exponent (α) of the 440 nm and 675 nm pair of wavelength [60].

$$AOD_{550nm} = AOD_{500nm} \times \left(\frac{550}{500}\right)^{-\alpha} \tag{1}$$

Today's alarming rate of climate change is based on alterations in land use and land cover [61]. Considering its major contribution to climate change, habitat loss, biodiversity loss, and improving human living standards, it is the most pressing issue within environmental assessment [62,63]. The environmental changes are directly linked with land use and land cover modifications that affect the soil moisture or the atmospheric heat budget. These are the two major constituents of a region's climate [64]. To plan for sustainable economic growth, land use planners must carefully consider the adverse effects of land use changes on the environment. To assess the environmental effects of LULC change, the International Geosphere and Biosphere Program (IGBP) and the International Human Dimensions Program (IHDP) collaborated and recommended research on LULC change consequences [63,65]. The LULC patterns were mapped using MODIS LULC product collection six level-3 (MCD12Q1) at 500 m spatial resolution obtained between 2010 and 2019. The data was accessed from the Data Access Center of LAADS [66]. Terra and Aqua data have been used to generate MODIS's Land Use Land Cover (LULC) product, which includes multiple classification schemes to describe land cover attributes. A supervised

decision tree classification method has been used in the International Geosphere-Biosphere Programme (IGBP) to define land cover classes (17), the classes of natural vegetation (11), classes with modified and mixed land (3), and classes of non-vegetated land (3), respectively. The Land _Cover_Type_1 SDS was used in this study since it provides data with the IGBP classification scheme. The data is divided into three basic classifications of land use land cover in this scheme: (1) Grassland, (2) Cropland, and (3) Built-up.

MODIS vegetation indices enable the comparison of vegetation conditions across geographies and over time. The MODIS daily vegetation indices are calculated using a combination of blue, red, and near-infrared reflectance. NDVI, a measure of the normalized difference vegetation index (NDVI), provides information regarding green biomass and vegetation growth status. This is typically used to monitor vegetation cover and type. This technique can reduce or eliminate the negative influence of error during instrument calibration, radiation present in the atmosphere, topography, and cloud cover when quantifying vegetation. Research on urban climate uses this method widely. The visible and near-infrared reflectance bands are used to derive this index:

$$NDVI = \frac{\rho NIR - \rho Red}{\rho NIR + \rho Red} \qquad (2)$$

where ρRed represents the reflectance value of band 1 and ρNIR band 2 for the MODIS satellite image.

The values of NDVI are between −1 to 1, positive NDVI indicates vegetation, and negative NDVI indicates non-vegetated surfaces. MODIS now has a new Enhanced Vegetation Index (EVI) product that reduces canopy background changes while maintaining sensitivity in dense vegetation. The EVI also utilizes the blue band's surface reflectance to eliminate smoke residuals and the sub-pixel of thin clouds in the atmosphere.

$$EVI = G \left( \frac{\rho NIR - \rho Red}{\rho NIR + C1 \times Red - C2\, Blue + L} \right) \qquad (3)$$

The atmospherically corrected bi-directional surface reflectance is used for calculating the MODIS-derived NDVI and EVI. These are also free from the errors associated with water, clouds, heavy aerosols, and cloud shadows, as they are already masked for such measures. The surface reflectance data included with the NDVI MODIS package was used to create the NDBI and SAVI indices. To measure the density of built-up areas and their degree of development, the Normalized Difference Built-up Index (NDBI) was widely used. It can be used to identify the appropriate threshold value for a particular study area to differentiate the detailed characteristics of LULC. The NDBI was more appropriate for quantitatively identification of built-up abundance and spatial variation than NDVI in some previous studies [67]. The value ranges from −1 to 1, where a higher value denotes a higher building density. It involves the following formula:

$$NDBI = \left( \frac{\rho MIR - \rho NIR}{\rho MIR + \rho NIR} \right) \qquad (4)$$

The mid-infrared and near-infrared reflectance values are denoted ρMIR and ρNIR, respectively. The study also used an index that expresses the soil information effectively, the Soil Adjusted Vegetation Index (SAVI), which can be derived from soil information by adding soil parameters together and can be derived by:

$$SAVI = (1 + L) \left( \frac{\rho NIR - \rho Red}{\rho NIR + \rho Red + L} \right) \qquad (5)$$

where L is the coefficient of (L = 0.5, 1.5).

The 16 days composite data of the level-3 global product with a projection of Sinusoidal (MOD13A2, MYD13A2) and 1 km spatial resolution is provided by MODIS. The data can be accessed through LAADS [66]. The 1_km_16_days_NDVI and 1_km_16_days_EVI have

been used to generate the NDVI and EVI maps for 2019, while 1_km_16_days_red_reflectance, 1_km_16_days_NIR_reflectance, and 1_km_16_days_MIR_reflectance have been used to calculate SAVI and NDBI.

**Table 1.** Information about the data used in the study.

| Data Description | Site | Duration | Sites to Download Data |
|---|---|---|---|
| AERONET (Version 3 Level 2 Aerosol Optical Depth at 500 nm) | Amity University | 2010, 2016, 2017, and 2018 | http://aeronet.gsfc.nasa.gov/ [68] |
| MCD19_A2 (AOD at 1 km) | Gual Pahari | 2017, 2018, 2019 | |
| MOD13A2 (16 days Terra composite of NDVI, EVI, Red, NIR, MIR reflectance at 1 km) | Gautam Buddha Nagar, Faridabad, Gurugram, Ghaziabad | 2010–2019 | https://ladsweb.modaps.eosdis.nasa.gov/ [66] |
| MYD13A2 (16 days Aqua composite of NDVI, EVI, Red, NIR, MIR reflectance at 1 km) | | 2019 | |
| MCD12Q1 (Land Cover type 1 at 500 m) | | 2010–2019 | |

We have analyzed point-based collocation and 3 × 3 pixels centered at each AERONET site in the validation part. The AERONET and MODIS provide different types of AOD measurements. The former delivers point measurement with a high temporal resolution, and the latter provides spatial measurement across the satellite overpass (Terra: 10:30, Aqua: 14:30) twice daily. Therefore, for matching the pixel value of $AOD_{MAIAC}$ with point-based $AOD_{AERONET}$ measurements, it is necessary to perform the averaging of (a) the $AOD_{AERONET}$ with the time of satellite overpass and (b) the $AOD_{MAIAC}$ taking a spatial window of 3 × 3 pixels centering the AERONET sites for coverage of the different type of aerosols and various landmasses [69].

A few statistics were utilized to calculate the retrieval accuracy of the MAIAC algorithms in this study. $AOD_{MAIAC}$ was validated for 2010, 2016, 2017, 2018, and 2019, as there is an absence of AERONET stations and minimal observations on the ground. We have calculated match-ups (N), correlation coefficients (R), and expected error (EE) as part of the statistical validation [20]. The equation used for EE is described in Equation (6). Expected error envelopes have evaluated the algorithm's performance with EE definition of the MAIAC algorithm $\pm (0.05 + 15\% \times AOD_{AERONET})$.

$$EE_{MAIAC} = \pm(0.05 + 0.15AOD_{AERONET}) \tag{6}$$

The impact of LULC on AOD has been assessed using the MODIS land cover product (500 m) and indices (1 km) after validation. The flowchart of the methodology is provided in Figure 2. The data for NDVI, EVI, and surface reflectance for 2019 have been downloaded. The sub-datasets were extracted and reprojected from the product files for further processing. The Area of Interest (AOI) has been clipped from the image for all the products. The data was converted to binary representation using division with a scale factor of 10,000. The already processed MODIS products were used for NDVI (Equation (2)) and EVI (Equation (3)). In contrast, the surface reflectance bands of MIR, NIR, and red were used for NDBI (Equation (4)) and SAVI (Equation (5)). MODIS MAIAC AOD has been separated for Terra and Aqua analysis using MATLAB to explore the link between Aqua and Terra with various land uses. To match the resolution of $AOD_{MAIAC}$ and LULC-derived indices, the resampling has been done to grids of size 1 km x 1 km using the Fishnet tool of ArcGIS10.2 software. Approximately 5035 grids were found in the region. Every grid was assessed for its average AOD, NDBI, NDVI, EVI, SAVI, and area fraction for several LULC categories, including cropland, built-up, and grassland. A Pearson correlation analysis was conducted using Statistica based on the AOD and the LULC-related metrics.

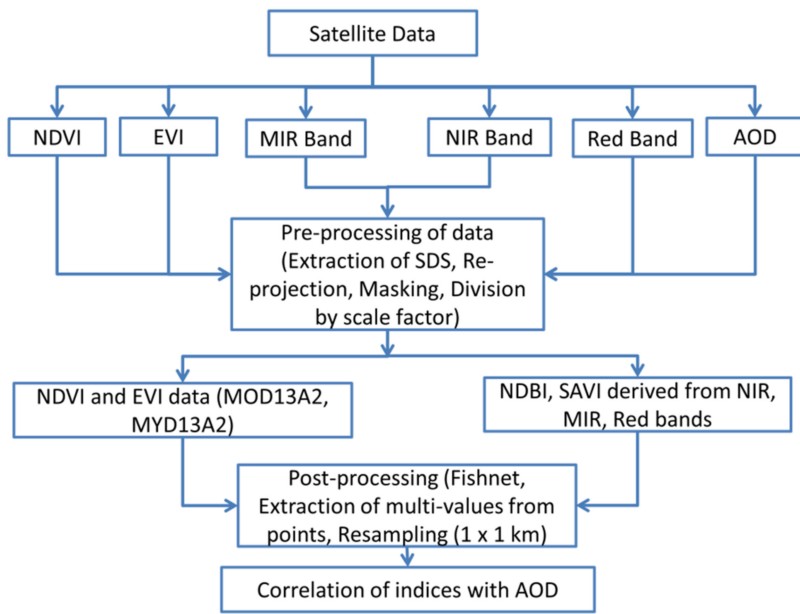

**Figure 2.** Flowchart of Methodology.

## 4. Results

### 4.1. Validation of AODMAIAC Using AODAERONET

The validation of $AOD_{MAIAC}$ with $AOD_{AERONET}$ has been attempted in the current study. In bin validation, the AOD values have been divided into two bins: AOD ≤ 0.5 and 0.5 < AOD ≤ 1, to know the magnitude of AOD for expected error. Approximately 93% of AOD values at Amity University and 79% at Gual Pahari fall inside the EE. This demonstrates that the AERONET AOD observations and MAIAC AOD values within 0.5 to 1.0 are more closely aligned. The underestimation also decreased in the present AOD range of 0.5 to 1.0. Considering the evaluation of the $AOD_{MAIAC}$ product with AERONET observations, it can be noted that MAIAC has outperformed the current area with a higher correlation with the ground measurements. $AOD_{MAIAC}$ has presented a better correlation with $AOD_{AERONET}$ for Amity University (0.86) than Gual Pahari (0.73). In general, the collective correlation coefficient for both the stations of Gurugram is 0.81, and RMSE is 0.16, with total match-up points for both stations being 105. Based on the analysis, the $AOD_{MAIAC}$ is more efficient than other traditional algorithms for some areas of the NCR; as the value of AOD becomes greater than 0.5, the bias increases. The result demonstrates that as the magnitude of AOD increases, the uncertainty of the MAIAC algorithm also increases.

Nearly 79% and 74% of the total $AOD_{MAIAC}$ retrievals lie within the EE envelope for point-based validation at Amity University and Gual Pahari, respectively. Compared to Gual Pahari, Amity University has more points within the expected Error (Table 2). Moreover, low underestimation has been observed at the Amity University site. Such observation shows that the MAIAC algorithm outperformed traditional algorithms (DB, DT) [32].

**Table 2.** Expected error envelope of $AOD_{MAIAC}$ in AOD bins of <0.5 and 0.5–1.0 (a) Amity University (b) Gual Pahari.

|  | EE | <0.5 | 0.5–1.0 |  |
|---|---|---|---|---|
| Amity University | % within | 70 | 92.86 | N = 105 |
|  | % below | 25 | 7.14 | R = 0.81 |
| Gual Pahari | % within | 65 | 78.79 | RMSE = 0.16 |
|  | % below | 35 | 21.21 |  |

### 4.2. Spatio-Temporal Variations of AOD

Spatial variation of AOD from 2010–2019 has depicted high AOD during 2015, 2016, and 2018 (Figure 3), while in 2019, AOD decreased, which could be attributed to the government policies toward pollution management in the study area. From 2010–2019, Ghaziabad and Noida had the highest AOD values, while fluctuations were observed in Faridabad and Gurugram. It was in 2018 that AOD reached its highest level. The high population density and the increased anthropogenic activities may cause high aerosol loading in the current study area [70].

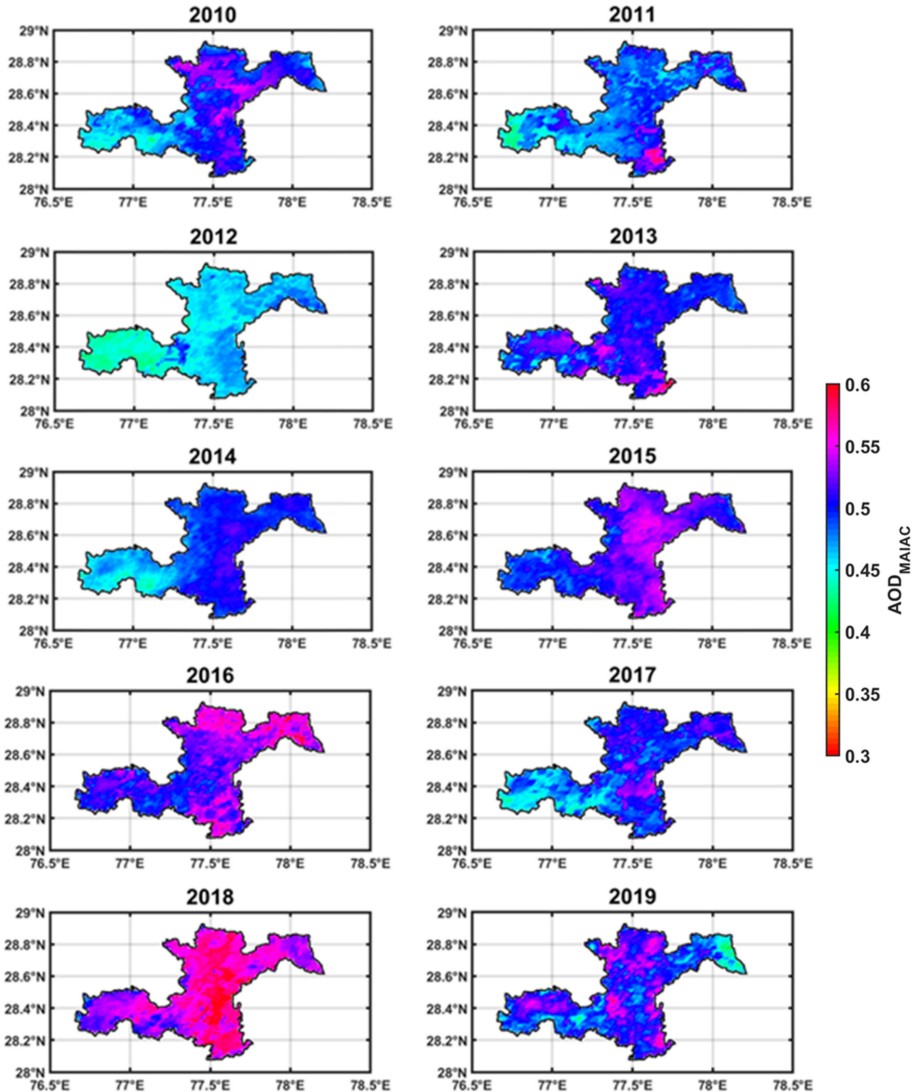

**Figure 3.** Spatial distribution of annual means of AOD$_{MAIAC}$ retrieved over Gual Pahari and Amity University from 2010 to 2019.

NCR has seen incredible spatial growth over the past decade, with a 62.5% urbanization level in 2011, and is expected to reach 71% by 2021. The reason for this rapid growth is the various quantity increase in the number of vehicles and the growth of industrial hubs [71,72].

In the monsoon season, north-western India plays a crucial role in aerosol loading through fertilizer and traditional cultivation [73,74]. Moreover, monsoonal rainfall adds little moisture to the air, which also triggers aerosol concentration during this season. An in-depth analysis of driving forces is required to determine which factor is responsible

for the significant increase in AOD. Therefore, it will be included in the future scope of the study.

### 4.3. Spatio-Temporal Variations of LULC and Its Impact on AOD

The most direct link between humans and nature is land use. As a result of our interactions with nature and the environment, LULC patterns provide a record of how and what we interact with. LULC types were categorized according to the IGBP classification of the MODIS LULC data file: cropland, built-up, and grassland throughout the study zone. Figure 4a depicts the spatial distribution of LULC in 2010, and Figure 4b for 2019. Table 3 exhibits the change in area by each LULC category from 2010 to 2019. The cropland region was estimated at 4442.31 km$^2$, accounting for 81.19% of the total. The cropland regions were mostly found outside of the city's center. Building sites were mainly located in the city center or along key suburban routes, accounting for 15.33% of the total (839.09 km$^2$). As shown in Table 3, the area of built-up land in urban areas has risen from 737.98 km$^2$ to 839.09 km$^2$, representing a growth rate of 12.05%. Grassland comprised 3.46% of the study area, which was 189.75 km$^2$. A significant decline in cropland has also been observed, as cropland occupies more than 40% of the entire area. Several of them are scattered throughout the area. Approximately 198.12 km$^2$ of cropland have been converted into built-up areas or grasslands. From Table 3 it can be analyzed that the built-up area has increased from 2010 to 2019.

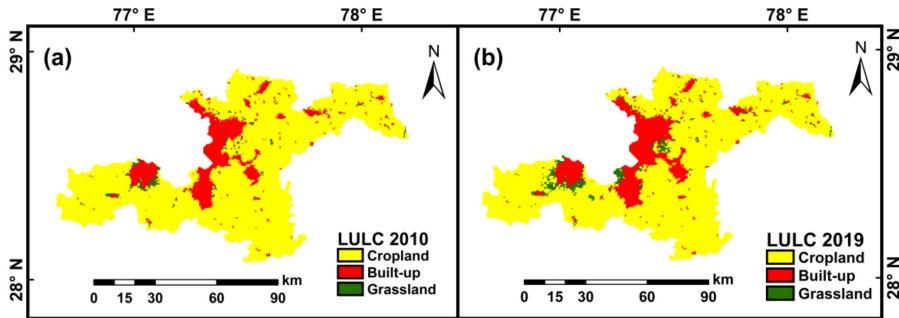

**Figure 4.** The Land use land cover distribution depicts three main classes: Cropland, Built-up, and Grassland: (**a**) 2010 and (**b**) 2019.

**Table 3.** Percentage change in the area of LULC classes from 2010–2019.

| LULC Class | Percentage Change (%) | | | | | | | | | Percentage Change in a Decade (2010–2019) |
|---|---|---|---|---|---|---|---|---|---|---|
| | 2011 | 2012 | 2013 | 2014 | 2015 | 2016 | 2017 | 2018 | 2019 | |
| Cropland | −0.69 | −0.42 | −0.55 | −0.88 | 0.12 | 0.26 | 0.15 | −0.05 | −2.35 | −4.46 |
| Built-up | 1.91 | 1.10 | 1.58 | 1.42 | 0.46 | 0.54 | 0.40 | 0.56 | 4.68 | 12.05 |
| Grassland | 15.79 | 8.88 | 9.77 | 17.57 | −5.87 | −11.90 | −7.94 | −1.88 | 34.27 | 51.13 |

Figures 5–8 depict the spatial distribution of various vegetation and built-up indices (NDVI, EVI, SAVI, and NDBI) derived from 16 days of Aqua and Terra composite. The NDBI value range (−0.44 to 0.73) is the same for NDBI derived from both Aqua and Terra composites. However, the difference in the range of index values is evident in vegetation indices derived from Aqua ($−0.18 \leq$ NDVI $\leq 0.78$) ($−0.14 \leq$ EVI $\leq 0.58$) and Terra ($−0.14 \leq$ NDVI $\leq 0.86$) ($−0.11 \leq$ EVI $\leq 0.62$) composites. Based on the analysis of Figures 5–8, it can be seen that between 2010 and 2019, the urban area grew from the city's core to the region's outskirts. Additionally, from 2010 to 2019, there was a decline in agriculture.

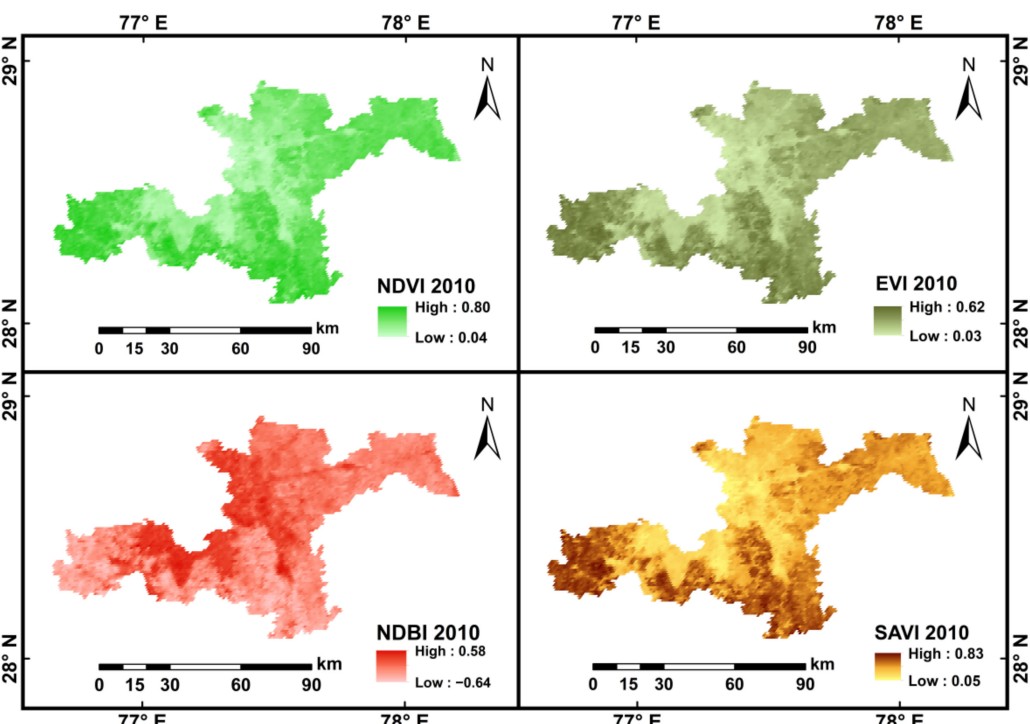

**Figure 5.** Maps of NDVI, NDBI, EVI, and SAVI of Aqua MODIS 2010.

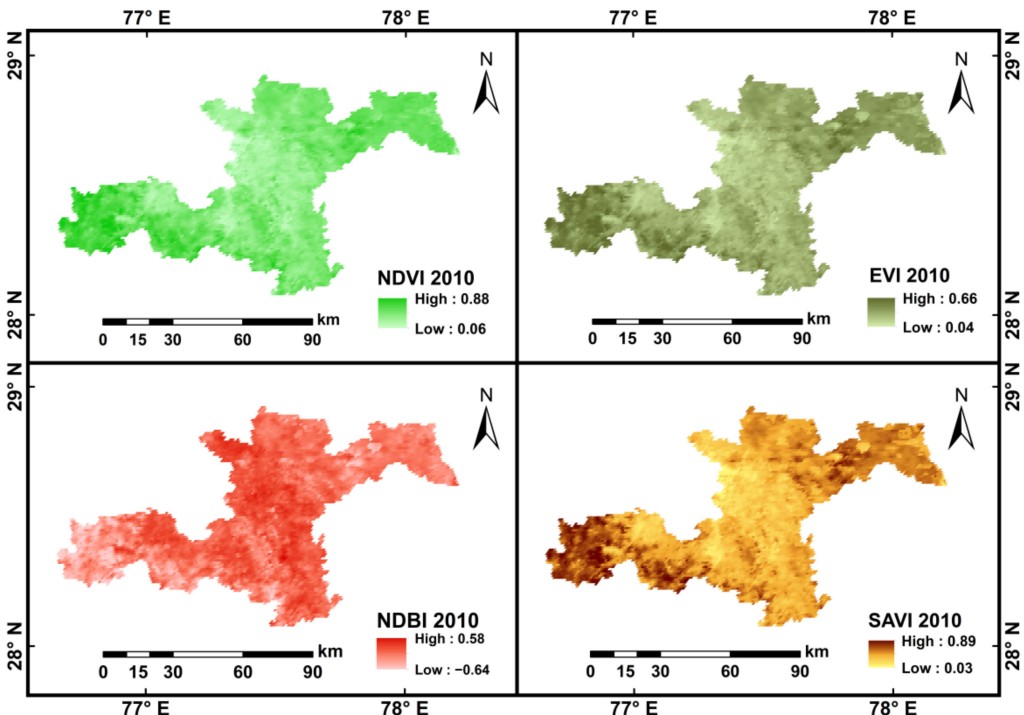

**Figure 6.** Maps of NDVI, NDBI, EVI and SAVI of Terra MODIS 2010.

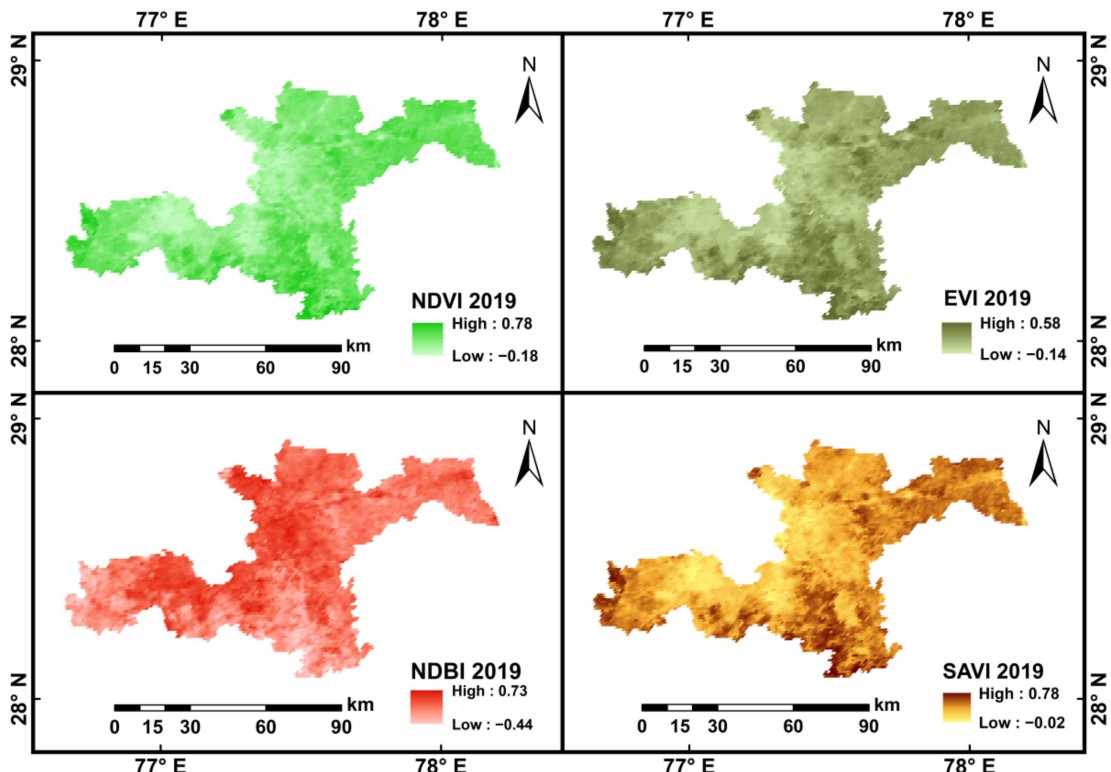

**Figure 7.** Maps of NDVI, NDBI, EVI, and SAVI of Aqua MODIS 2019.

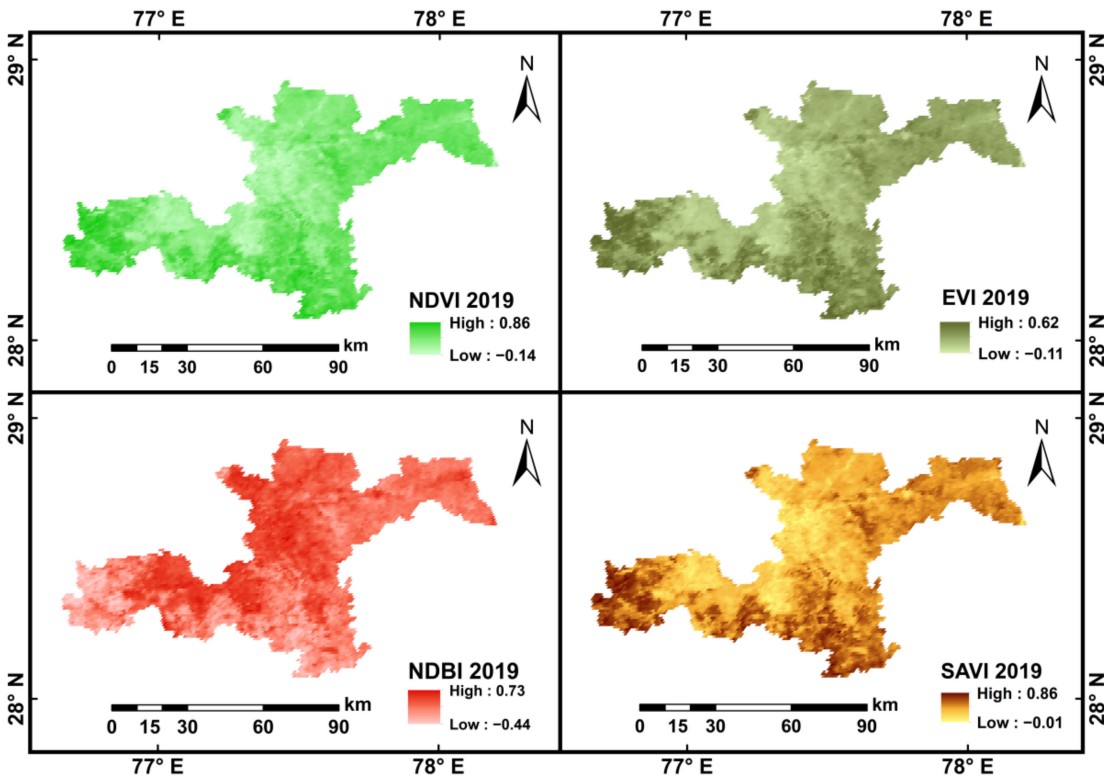

**Figure 8.** Maps of NDVI, NDBI, EVI and SAVI of Terra MODIS 2019.

More vegetation is usually indicated by higher and positive NDVI, EVI, and SAVI values. Negative and lower vegetation indices are associated with urban and rural development. The city area of Faridabad, Gurugram, Ghaziabad, and Gautam Buddha Nagar are

associated with high positive NDBI values because of the dense built-up region. Besides the city area, NDBI is also higher in rural areas where vegetation is absent, i.e., barren land, open/protected areas, and grazing land [51,52,75]. In adjacent suburbs where plantation or native farmland predominated, NDVI, EVI, and SAVI values are positive and higher. A progressive increase in positive values of vegetation indices has been observed from sparse grassland to dense cropland.

Figure 7 depicts the lowest values of NDVI and EVI as −0.18 and −0.14, respectively, while the lowest value of SAVI is −0.02. Lowering the index value or wider range of index value in the negative region for NDVI and EVI refers to a large proportion of the area under no vegetation. However, the narrow range of SAVI in the negative region depicts a lesser proportion of the area with no vegetation. It indicates that significant parts of the study area with less dense/thorny bushes in the rural zone and sustainably designed built-up areas interspaced with plantations in the urban zone have been detected by SAVI, unlike NDVI and EVI. Hence, SAVI is proven to be a more powerful tool to detect areas with fragmented/less dense vegetation where reflectance from underlying soil is combined with the reflectance of vegetation. It can also be concluded that combined analysis by NDVI and SAVI can be used to distinguish dense and light vegetation areas. Similar observations can also be detected in Figure 8.

To examine the deviation of AOD with the spatial variation of LULC, values of minimum, maximum, average, and standard deviation (SD) of AOD in each LULC category have been calculated (Table 4). Average AOD in built-up regions was the highest, with 0.70 for aqua and 0.68 for terra, followed by Grassland (0.69, 0.66) and Cropland (0.67, 0.65). The abundance of anthropogenic activities and traffic density in urban areas is the root cause of high aerosol concentration. AOD values were generally lower for areas covered by matured crops and grasslands. It is implied that heavily vegetated areas produce a cleaner environment. However, with the maximum area coverage in the study area, cropland had the widest range of AOD values (0.54–0.85 (aqua) and 0.53–0.82 (terra)) with significant discrepancy depicted through the highest SD (0.05 (aqua) and 0.04 (terra)) among all LULC types. In the study area, considerable use of fertilizer in agriculture to facilitate crop growth and, from time to time, stubble burring could also increase AOD over the cropland [76].

**Table 4.** Statistical analysis of Aqua and Terra AOD for LULC classes in the current region.

|  | Cropland | | Built-Up | | Grassland | |
|---|---|---|---|---|---|---|
|  | *Aqua* | *Terra* | *Aqua* | *Terra* | *Aqua* | *Terra* |
| *Mean* | 0.67 | 0.65 | 0.70 | 0.68 | 0.69 | 0.66 |
| *S.D.* | 0.05 | 0.04 | 0.04 | 0.03 | 0.04 | 0.04 |
| *Min* | 0.54 | 0.53 | 0.59 | 0.56 | 0.59 | 0.56 |
| *Max* | 0.85 | 0.82 | 0.79 | 0.77 | 0.80 | 0.76 |

To investigate the influence of the different proportions of built-up and vegetation over aerosol concentrations, a correlation analysis has been performed between the spectral indices indicator of the density of vegetation and built-up, and AOD, an indicator of aerosol concentration. Table 5 provides coefficients of correlation generated between the AOD and spectral indices. At the significance level of 0.01, all the selected indices were strongly related to AODs, except for NDBI. It was found that indicators associated with urbanization, such as NDBI, correlated positively with AOD, as expected. On the other hand, the correlation coefficient was only 0.35. The NDVI and EVI were negatively correlated with AOD with the value of −0.24, −0.15, which shows that the vegetation has a purification impact on AOD. The build-up increase has also raised AOD values, according to the positive NDBI correlation (0.35). The NDVI, NDBI, EVI, SAVI, and AOD correlation coefficients were not high enough, indicating that there may not be a clear or continuous association on the city scale and that indexes alone are insufficient to explain the variance

in AOD across the research region fully. Consequently, a suitable landscape context should be used for analysis to support further research.

**Table 5.** The correlation coefficient of AOD with NDVI, NDBI, SAVI, and EVI.

|  | NDVI | NDBI | SAVI | EVI |
|---|---|---|---|---|
| R | −0.24 | 0.35 | 0.27 | −0.15 |

AOD values were negatively related to the variables associated with vegetation, namely NDVI and EVI, and positively correlated with the soil coverage index, SAVI. The analysis of Table 5 demonstrated that the NDVI, the EVI, and the AOD are weakly related. A vegetation canopy can absorb atmospheric particles, particularly in dusty conditions. It has been demonstrated that vegetation can effectively remove aerosols from the air due to its adsorption and removal capabilities. The current study zone is semi-arid, with rocky and barren lands contributing to higher AOD. Here the SAVI positive correlation justifies that in the current study area, the soil plays an important role, and further classification is required to understand the underlying facts of the SAVI and AOD relationship.

LULC types had a considerable impact on AOD levels, according to the above findings. To better understand how LULC affects the distribution of AOD, scatter plots with densities were displayed to illustrate correlations of NDVI, NDBI, EVI, SAVI, and AOD. A quantitative link between AOD and NDVI values is shown in Figures 9–12. Cropland dominates the current research region; built-up areas are concentrated in the center and sparsely distributed. Grassland is completely dispersed, as shown in Figures 9–12.

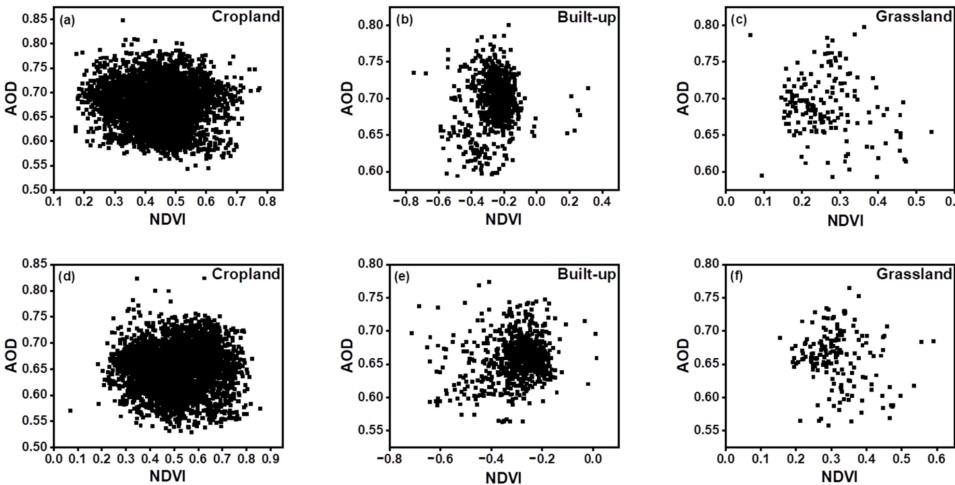

**Figure 9.** Scatterplot depicting the relationship between NDVI and AOD (Aqua (**a–c**), & Terra (**d–f**)) values for different LULC classes.

It was discovered that AOD and NDBI have a weak but positive association in scatterplots (Figure 10), implying that while NDBI alone may not adequately explain aerosol spatial variation, it can influence the growth of AOD. Natural surroundings altered the effect of built-up areas on aerosol concentration, thus weakening the link between them. The association of AOD with EVI is depicted in Figure 11. The following diagrams show the vegetated region that was not included in the NDVI analysis.

Figure 11 shows a low and negative regional association between aerosol amount and improved vegetation bodies in the scatterplots between AOD and EVI. There was, however, greater consistency between AOD and EVI in agricultural areas compared to grassland areas. Because of the negative association between these two variables, increasing cropland and grassland acreage could reduce aerosol deposition.

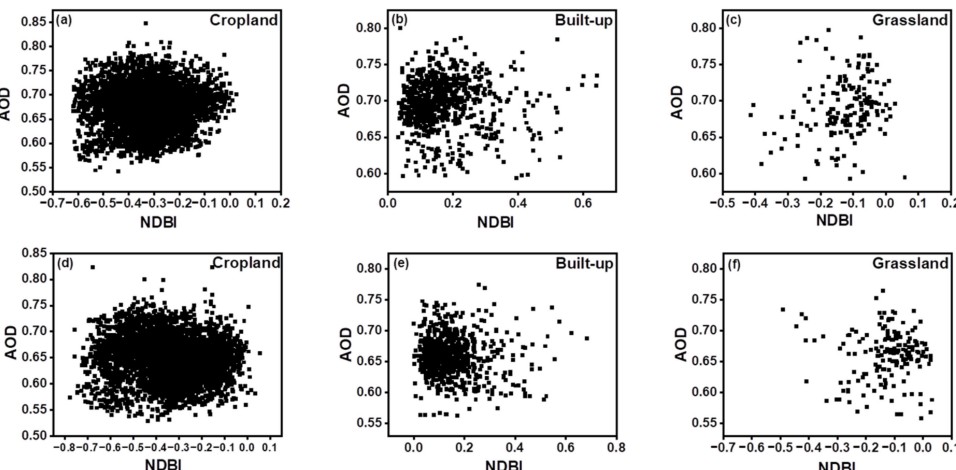

**Figure 10.** Scatterplot depicting the relationship between NDBI and AOD (Aqua (**a**–**c**), & Terra (**d**–**f**)) values for different LULC classes.

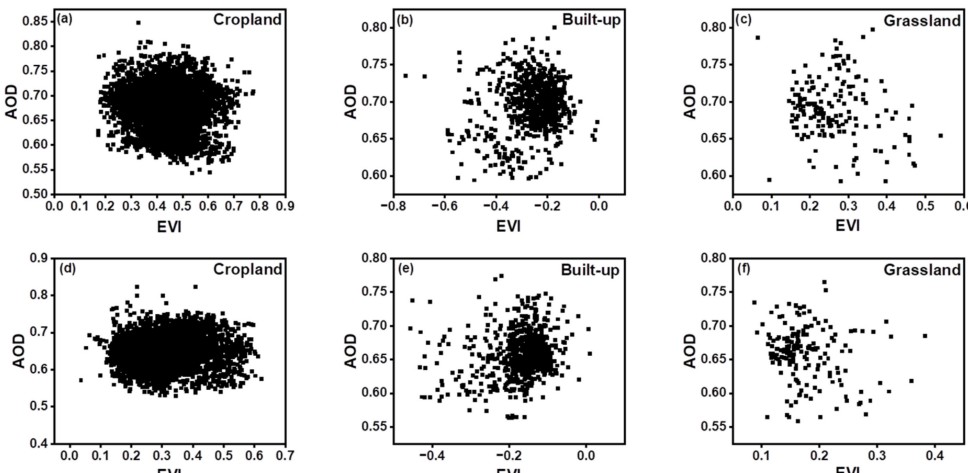

**Figure 11.** Scatterplot depicting the relationship between EVI and AOD (Aqua (**a**–**c**), & Terra (**d**–**f**)) values for different LULC classes.

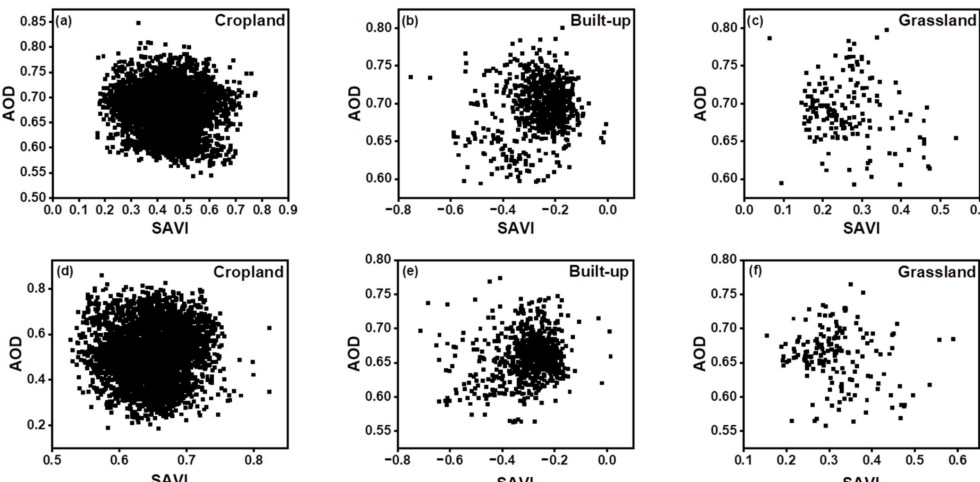

**Figure 12.** Scatterplot depicting the relationship between SAVI and AOD (Aqua (**a**–**c**), & Terra (**d**–**f**)) values for different LULC classes.

The quantitative link between AOD and SAVI values is depicted in Figure 10. Because the current study location is in the semi-arid zone and contains numerous barren and rocky parts, the impact of this area on AOD and the current index has been investigated. Figure 12 shows a weak and positive spatial association between aerosol quantity and soil-adjusted vegetation index in the scatterplots between AOD and SAVI.

However, in agriculture and built-up regions, AOD values were more consistent with SAVI than in grassland. Because these two variables have a positive association, increasing the rocky and barren land area could promote aerosol deposition.

The current study area is located within the arid zone of the Koppen climate zone classification, which features shrublands and barren (sand, gravel, drought-resistant plants, among others) characteristics of arid climates [77]. Therefore, a weak correlation was established between the indices drawn for vegetation and AOD.

## 5. Discussion

The MODIS aerosol products with a spatial resolution of 1 km × 1 km were used to map the AOD distribution. LULC classification was determined using a MODIS with a spatial resolution of 500 m by 500 m. This database analyzed variability in AOD values across different LULC categories and correlations between AOD and LULC-related variables. High-resolution MAIAC AOD has given a new perspective on city-level aerosol analysis. MAIAC, the recent algorithm with high-resolution data of 1 km, assists in determining air quality in highly urbanized areas and thus identifies the pollution hotspot. The spatial complexity of AOD distribution must be detected to understand the impact of heterogeneity in the landscape composition of the region on aerosol pollution. In the current study, MAIAC-derived AOD was validated against AERONET-derived AOD, with more than 70% of retrievals occurring within the EE. Furthermore, the study's results showed an increasing trend in AOD from 2010–2019, with 2018 marking its highest level. In various other studies, the same trend was observed in the increment in AOD [32,34,78].

The main reason for atmospheric aerosol concentration was the anthropogenic emission of air pollutants in urban areas [79]. An unexpected urbanization level of 62.5% in the National Capital Region in 2011 may explain the increase in the trend of AOD and pollution. The percentage is expected to increase to 71% by 2021. This results in the vehicle's multiple growths, the development of various industrial hubs (large, medium, and small-scale industries), and the construction of brick kilns [71,72,80]. The rural area of north-western India, aside from the urban area, contributes significantly to aerosol loading, especially during the monsoon season, due to intensive conventional cultivation and fertilizer use [73,74].

To understand how urbanization and other prominent types of LULC are changing in the current study area, we estimated the LULC change with MOD12Q1 land cover product in the current study area. In this study, we have analyzed that urban built-up has increased by 12.05%. In contrast, a decrease in cropland by 4% has been seen (Figure 3), which is also confirmed by the cited studies that there is an increase in built-up land and a decrease in cropland between 2000 to 2020 [13,17] as the LULC alone can only aid in a qualitative study of AOD and LULC correlations and cannot satisfy the needs of the study. Therefore, the indices such as NDVI, NDBI, and SAVI have been calculated to aid a quantitative investigation.

To determine the most significant LULC factor in urbanized regions, it is necessary to investigate LULC in such places. The LULC structure contains information on the landscape's variety. Remote sensing and in-situ data have been used to find the relationship between the LULC pattern and AOD distribution and pattern by studying the spatial distribution of vegetation index, built-up index, and soil index. There is an interrelated relationship between LULC change and local climate change. Climate change can result in changes in LULC and vegetation cover. In order to fully understand the relationship between LULC and the local climate, continued scientific research is needed. The increase in built-up areas has a maximum effect on pollution in the current study area. This implies that AOD and LULC have a cause-and-effect relationship. The positive correlation between

AOD and NDBI, SAVI, in semi-arid regions can be explained by increased built-up and soil particles. The overall association was weak, indicating that the LULC factors alone should not be utilized for further pollution calculations.

As far as we know, in Indian regions, there was no such research on such a perspective of correlation between AOD and LULC-derived parameters. Therefore, the authors have tried to compare the results with the studies conducted in Chinese regions. Various studies in different polluted areas of China like Wuhan, Beijing, and Shijiazhuang ([43,81–84]) with different satellite-derived data (MODIS, Landsat, Sentinel-2, GlobeLand30, and ASTER) have shown that AOD was positively correlated with built-up and negatively with vegetation indices. The LULC and its structure contribute to aerosols and their variation. The findings suggested that vegetation is of great importance for decreasing AOD; urbanization increases aerosol pollution on a city scale instead. The urban areas should have dense and porous vegetation to balance the deposition and dispersion of pollutants. The purification effect of vegetation is also affected by several parameters, such as ventilation, topography, and pollutant concentration. Our results have also shown similar findings to the development of built-up leads to increased air pollution. The NDBI and AOD have shown a positive correlation, and NDVI, EVI, and AOD have a negative correlation supporting the purification of AOD by increasing the vegetation by planting more trees.

These studies found high AOD values in areas with highly sparse vegetation and urban built-up surfaces. Similar observations of high mean AOD value were found for built-up in the current study. The current study found that SAVI correlated positively with AOD, indicating a need to classify the LULC types further and study the landscape context to understand better the impact of SAVI on AOD in the current semi-arid region. SAVI is a better parameter to be included in the modeling of $PM_{2.5}$ for semi-arid areas with sparse vegetation or barren land and to understand the impact of soil on air pollution. The following limitations of the study can be further improved in the future: (a) Irregular ground-based aerosol measurements due to limited AERONET stations; (b) The study's period was too short to determine LULC changes, which could be improved by considering long-term data; (c) The coarse resolution data for classification of LULC types. Future studies may focus on the detailed classification and its effect on aerosols and air pollution.

However, the results of this study suggest that good network AERONET sites must be developed to record the long-term AODs in specific study areas systematically. Based on our knowledge and research of the situation, there are no reports on the correlation of LULC with AOD in the current study region. This study also found that LULC had a variable effect on AOD concentration depending on the land cover. Overall, there was a weak association, indicating that the LULC factors alone should not be utilized to calculate pollutants further. However, AOD is generally region-dependent, with climatic conditions having the greatest impact. As a result, the LULC and indicators can play an important role in future pollution research. This study can help determine more parameters while modeling $PM_{2.5}$. Furthermore, it may be possible to control future scenarios of pollution at local levels as well as to implement specific mitigation measures at the local level to achieve sustainable development goals by using this information. The environmental authorities, urban planners, urban ecologists, and climatologists can use the study's results of the interaction between human activities and environmental quality and control air quality and land management problems at the city scale.

## 6. Conclusions

In most previous studies, only NDVI or NDBI has been used as a related parameter. In the semi-arid study regions, variables such as SAVI and EVI should also be considered important parameters. Therefore, this study aimed to explore the impact of LULC-derived indices on AOD in parts of Delhi, NCR, as well as provide a new perspective on how aerosol variation responds to land use patterns at the regional scale. The 77% of MAIAC retrieved AOD lies within the expected error with nil overestimation. MAIAC retrieved AOD is within the limit of EE and represents a robust correlation for rural and semi-urban sites,

i.e., Gual Pahari and Amity University, respectively. LULC has significantly changed from 2010–2019 in the current region, showing rapid increases in built-up areas and grassland and decreased cropland. The total 12% of built-up and 51% of grassland area increased in 2010–2019, whereas the cropland decreased by approximately 4%.

The highest mean AOD value was found in built-up areas (0.70), followed by grasslands (0.69) and croplands (0.68). According to the study's results, AOD was positively correlated with NDBI and SAVI at the significance levels of 0.01 and 0.05, respectively. The results indicate that urban development also gives rise to pollution concentration, whereas vegetation has a purification effect. The positive correlation between AOD and built-up areas indicates that urban development has increased aerosols and air pollution levels on a mesoscale. The soil particles may also contribute to the current semi-arid region, so it is necessary to study the landscape context. In semi-arid regions, soil or dust particles are the main pollutants in aerosol concentration. For semi-arid regions like the current study area, the SAVI can be an important factor for modeling communities as an influential parameter. There was a negative correlation between AOD and vegetated areas. To improve the efficiency of aerosol purification, vegetation coverage should be increased. Our results confirmed that LULC and its structure significantly affect aerosols and their variation. Planning and land use managers may use these findings to develop appropriate urban planning and land use management strategies. Moreover, by considering factors similar to SAVI and EVI, the modeling community can enhance models for assessing $PM_{2.5}$ in data-scarce regions. Future studies may focus on (a) the long-term evaluation of LULC, and AOD impacts should be studied; (b) many more parameters should be considered for further analysis; (c) high-resolution imagery should be used for LULC classification and could provide a better insight into various other land cover types, and (d) the comparison of different types of AOD classes in different types of LULC classes.

**Author Contributions:** Conceptualization, S.G.; Methodology, V.S.; software, V.S. and D.K.V.; validation, V.S. and S.G.; formal analysis, V.S.; investigation, V.S. and D.K.V.; resources, V.S.; data curation, V.S.; writing—original draft preparation, V.S.; writing—review and editing, S.G., S.S., D.K.V., A.K. and R.K.T.; visualization, V.S.; supervision, S.G., A.K. and S.S.; project administration, S.G.; funding acquisition, N.A.-A. and A.K. All authors have read and agreed to the published version of the manuscript.

**Funding:** This research received no external funding.

**Institutional Review Board Statement:** Not applicable.

**Informed Consent Statement:** Not applicable.

**Data Availability Statement:** Not applicable.

**Acknowledgments:** MODIS AOD products were available at Level-1 Atmosphere Archive & Distribution System (LAADS) (https://ladsweb.modaps.eosdis.nasa.gov/, accessed on 8 October 2022). The authors thank the NASA AERONET federation, AERONET scientific team, and principal investigators for establishing, maintaining the sites, and providing AERONET data at https://aeronet.gsfc.nasa.gov/, accessed on 8 October 2022). Alban Kuriqi is grateful for the Foundation for Science and Technology's support through funding UIDB/04625/2020 from the research unit CERIS.

**Conflicts of Interest:** The authors declare no conflict of interest.

## Abbreviations

| | |
|---|---|
| AERONET | Aerosol Robotic Network |
| AOD | Aerosol Optical Depth |
| EE | Expected Error |
| EVI | Enhanced Vegetation Index |
| GIS | Geographic Information System |
| IGBP | International Geosphere and Biosphere Program |
| IGP | Indo-Gangetic Palin |



| IHDP | International Human Dimensions Program |
| LULC | Land Use Land Cover |
| MAIAC | Multiangle Implementation of Atmospheric Correction |
| MODIS | Moderate Resolution Imaging Spectroradiometer |
| NCR | National Capital Region |
| NDBI | Normalized Difference Built-up Index |
| NDVI | Normalized Difference Vegetation Index |
| PM | Particulate Matter |
| RS | Remote Sensing |
| SAVI | Soil Adjusted Vegetation Index |
| SD | Standard Deviation |
| SEZ | Special Economic Zone |

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
