# Peer review of "Spatial Variation and Relation of Aerosol Optical Depth with LULC and Spectral Indices"

_atmosphere, doi:10.3390/atmos13121992_

Round 1

Reviewer 1 Report

The manuscript is good, the authors evaluate the ‘’ Spatial variation and relation of aerosol optical depth with LULC and spectral indices’’. It is an interesting and great contribution to the scientific community; however, the material method, discussion and references of the paper should be improved. Still there are many issues present in the manuscript which should be explained properly. The manuscript needs some minor revisions as given below:

·         The text of this paper in general needs a thorough review, as there are multiple spelling and grammatical errors. Many sentences do not mean any sense. Moreover, there are several sloppy errors that should be fixed.

·         Please add the method sentence in Abstract.

·         Write the main objective of this study in the points at the end of introduction.

·         The paper provides a good documentation. But it has little to do with LULC. The data series 2008-2020 is simply too short to say anything about LULC.

·         I suggest, add the flow chart of your research methodology. See suggested references.

·         Table 3, Add % of all years.

·         Fig. 4, 5, These images for the year of 2019, where is images of 2010

·         Where is Scatterplot depicting the relationship between NDVI and AOD ??

·         More research background and motivation should be added to the Introduction section. Although, I propose some new papers must be added in the reference list and text which will also help you to make it more intriguing such

https://doi.org/10.1038/s41598-022-17454-y

https://doi.org/10.3390/land11050595

·         Resolution of all figures should be improved.

·         In discussion section; Discussion: As per the instruction given by the journal “The findings and their implications should be discussed in the broadest context possible and the limitations of the work highlighted”.

·         Write main results and future recommendation in conclusion.

Overall, the study conducted is interesting but a minor revision of the entire manuscript is essentially required for publication in this journal. Hence, I recommend reconsideration after a minor revision of the manuscript.  

Author Response

Reviewer 1

The manuscript is good, the authors evaluate the " Spatial variation and relation of aerosol optical depth with LULC and spectral indices". It is an interesting and great contribution to the scientific community; however, the material method, discussion and references of the paper should be improved. Still there are many issues present in the manuscript which should be explained properly. The manuscript needs some minor revisions as given below: 

  • The text of this paper in general needs a thorough review, as there are multiple spelling and grammatical errors. Many sentences do not mean any sense. Moreover, there are several sloppy errors that should be fixed.

Reply: Per the reviewer's suggestions, the spelling and grammatical errors have been modified to the best of our knowledge. Kindly refer to the "Revised Manuscript".

  • Please add the method sentence in Abstract.

Reply: According to the reviewer's suggestions, the abstract has added the method sentence. "The current study used remote sensing and Geographical Information System (GIS) techniques to examine changes in LULC in the current study region over the ten years (2010-2019), as well as the relationship between LULC and AOD", has been added in the revised manuscript. Kindly refer to the "Abstract" in the "Revised Manuscript".

  • Write the main objective of this study in the points at the end of introduction. The paper provides a good documentation. But it has little to do with LULC. The data series 2008-2020 is simply too short to say anything about LULC.

Reply: According to the reviewer's suggestions, the study's main objectives have been mentioned in the points form at the end of the introduction.

"The main objectives of the study are:

  1. To study the spatial variation of AOD in the current study area.
  2. To analyze the change in LULC from 2010 to 2019.
  • To examine the correlation between AOD and LULC-derived indices."

Kindly refer to the last paragraph of the "Section 1: Introduction" in the "Revised Manuscript".

  • I suggest, add the flow chart of your research methodology. See suggested references.

Reply: Authors agree with reviewer's comment that the data series of LULC from 2010-2019 is too short of saying anything about LULC. The long-term data analysis for the current study area has been provided as a future recommendation. In the current study, the analysis of the relationship between LULC-derived indices and Aerosol Optical Depth (AOD) has been considered the main goal of this research, which will be used to help select potential parameters for PM2.5 prediction. A comparison of LULCs for 2010 and 2019 is provided to illustrate the changes in different classes of LULC for the current study area and to support the statements of an increase in the built-up areas and a decrease in cropland.

  • I suggest, add the flow chart of your research methodology. See suggested references.

Reply: According to the reviewer's suggestions, the methodology flowchart has been added to the Revised Manuscript. Kindly refer to "Figure 2" of the "Revised Manuscript."

  • Table 3, Add % of all years.

Reply: As per reviewer's suggestions, the percentage for all the years has been added in Table 3. Kindly refer to "Table 3" of the "Revised Manuscript".

  • 4, 5, These images for the year of 2019, where is images of 2010.

Reply: As per reviewer's suggestions, the images for the year 2010 have been added as Figure 5 and Figure 6. Kindly refer to the "Figure 5, 6" of the "Revised Manuscript".

  • Where is Scatterplot depicting the relationship between NDVI and AOD?

Reply: According to the reviewers' suggestions, the Scatterplot for NDVI and AOD has been provided in Figure 9. Kindly refer to the "Figure 9" in the "Revised Manuscript".

  • More research background and motivation should be added to the Introduction section. Although, I propose some new papers must be added in the reference list and text which will also help you to make it more intriguing such as: https://doi.org/10.1038/s41598-022-17454-y & https://doi.org/10.3390/land11050595

Reply: Thank you for your suggestions. As suggested, the Revised Manuscript has incorporated the research background and motivation.

"There is the creation of special economic zones (SEZ) in places like Noida and Gurugram, as well as the expansion of industrial enterprises and the financial and private sectors, are what are to blame for the increase in urbanization [13,17]. This has drawn a significant amount of labor force to the region, eventually contributing to population expansion and changes in the LULC pattern of the current research area during the previous 2010-2019. Rapid economic development triggers a noticeable change in LULC within a relatively short period in the study region. However, the research region includes the top 10 most polluted cities listed in the World Air Quality report [38, 39]. However, surprisingly, few studies have concentrated on the mesoscale or local level of the current research region [40, 41]. Therefore, the current study has thus attempted to provide a preliminary study in such a data-scarce zone."

Kindly refer to paragraph 5 of "Section 1: Introduction" in the "Revised Manuscript".

  • Resolution of all figures should be improved.

Reply: The resolution of all the images has been improved per the reviewer's suggestions. Kindly refer to the "Revised Manuscript".

  • In discussion section; Discussion: As per the instruction given by the journal "The findings and their implications should be discussed in the broadest context possible and the limitations of the work highlighted".

Reply: According to the reviewers' suggestions, the revision manuscript has improved the discussion part. Kindly refer to "Section 5: Discussion" in the "Revised Manuscript".

  • Write main results and future recommendation in conclusion.

Reply: As per the reviewer's suggestions, the conclusion has been modified by adding the main results and future recommendations. Kindly refer to "Section 6: Conclusion", in the "Revised Manuscript".

Overall, the study conducted is interesting but a minor revision of the entire manuscript is essentially required for publication in this journal. Hence, I recommend reconsideration after a minor revision of the manuscript.  

Reply: Thank you for your comprehensive suggestions and encouraging us. I hope your suggestion will improve the manuscript's quality.

Reviewer 2 Report

Please see the file attached for the comments.

Author Response

Reviewer 2

In the paper entitled “Spatial variation and relation of aerosol optical depth with LULC and spectral indices”, the authors examined the changes in LULC for the studied region, analyzed the decadal variation of AOD, and correlated AOD with LULC indices. Generally, this paper is well written, which favors a possible publication in the journal Atmosphere. However, following modifications need to be performed.

Content:

  • Equation (1) converts the AOD of 500nm to 550nm using Angstrom's equation, but does not indicate the value of the coefficient α.

Reply: As per the reviewer’s suggestions, the sentence has been modified in the Revised Manuscript. AODMAIAC is available at 550 nm and AODAERONET at 500 nm, so to compare the data, AODAERONET interpolating to 550 nm was done using Angstrom’s equation by using Angstrom Exponent (α) of the 440 nm and 675 nm pair of wavelengths [60].” Kindly refer to the “Section 3, paragraph no 2,” in the “Revised Manuscript”.

  • In Section 4.1, the AOD data of MAIAC is verified with AERONET ground observation data, but there are only two AERONET sites, both of which are located in Gurugram. Personally, I think that only the data of two stations are used to verify the remote sensing AOD data of four regions, which may lead to greater contingency.

Reply: The authors agree with the reviewer’s comment that only two AERONET stations were used in the current study as these were the only two stations available in the current study region. The authors have also discussed the current region in the data-scarce zone of Delhi, NCR. Through this study, authors have tried to grab the attention of regulating authorities to provide attention to such a data-scarce region, which is more polluted than Delhi.

  • MAIAC-AOD data include data from two satellites (Aqua and Terra), but they are compared together in table 2. Is that good?

Reply: The MAIAC AOD validation is not a major study part. Here, the validation has been shown to support a validated data used thought. Therefore, the validation is shown collectively, and the detailed analysis of Aqua and Terra is done in another study under process. For the correlation analysis, the MAIAC AOD data has been segregated to Aqua and Terra in the current study, as can be seen in Figure 9-12.

  • The relationship between AOD and NDVI, SAVI and other indexes is analyzed in Table 5. NDVI and EVI are negatively correlated, while NDBI and SAVI are positively correlated. Both SAVI and NDVI are vegetation indexes. The higher the value, the denser the vegetation coverage. But compared with NDVI, SAVI can remove the change of vegetation index caused by soil. However, in this paper, SAVI and AOD are positively correlated, which means the lusher the vegetation is, the more aerosols there are. This is somewhat inconsistent with the author's conclusion: "Here the SAVI positive correlation judges the fact that in the current study area, the soil is positively correlated to AOD".

Reply: Author’s agree with the reviewer’s statement that positive relation with SAVI means the lusher the vegetation is, the more aerosols there are. In the current research region, which is a semi-arid area, there are additional LULC types, such as fallow or barren land or rocky terrain, which contribute dust or soil particles to AOD. Therefore, it has been advised to use high-resolution data for LULC classification and landscape context analysis to uncover additional information about the SAVI connection for the current research region. The reviewer's suggestions were considered, and modification has been done regarding SAVI conclusions.

Format:

  •  

Reply: According to the reviewer’s suggestion MIR and NIR have been italicized, and Font in line 204 has been changed in the Revised Manuscript. “The mid-infrared and near-infrared reflectance values are denoted MIR and NIR, respectively.” Kindly refer to the “line number 204” in the “Revised Manuscript”.

  • The last row of Table 1 is on the next page, and the website in the last column is not vertically centered.

Reply: According to reviewer’s suggestion, the “Table 1” last row has been modified, and the website has been vertically centered in the Revised Manuscript. Kindly refer to the “Table 1” in the “Revised Manuscript”.

  • Line 230: “The equation used for EE is described in Eq. no.1.” But in fact, the EE calculation equation is in line 144, equation (1) is to convert the AOD of 500nm into the AOD of 550nm by angstrom equation.

Reply: According to the reviewer’s suggestion, the equation number has been changed in the Revised Manuscript.The equation used for EE is described in Eq. no. 6.” Kindly refer to the “line number 230” in the “Revised Manuscript”.

  • The secondary title has two 4.2.

Reply: The secondary title of 4.2 has been changed to 4.3. Kindly refer to the “section 4.3” in the “Revised Manuscript”.

  • There is no Figure 6.

Reply: According to the reviewer’s suggestions, Figure 6 has been added to the Revised Manuscript as Figure 9. Kindly refer to “Figure 9” in the “Revised Manuscript”.

  • Figure 2, 7, 8, 9, the font is too small.

Reply: According to reviewer’s suggestions, the font size of figures 3, 9-12 has been increased in the Revised Manuscript. Kindly refer to “Figure 3, 9-12” in the “Revised Manuscript”.

  • The latitude and longitude scales of Figure1, 3, 4 and 5 are in the Figure, but in Figure 2, it is out of the figure, which is not unified.

Reply: As per reviewer’s suggestions, the latitude and longitude of figures 3, 4-8 has been unified in the Revised Manuscript. Kindly refer to “Figure 3, 4-8” in the “Revised Manuscript”.

  • Figure 7, 8, 9 are scatterplots of two kinds of data (Terra and Aqua), AOD of three types of LULC (cropland, built - up, grassland) and three types of index (NDBI, EVI, SAVI), ordinate is AOD, the abscissa is three index value. I think the label of the abscissa should be NDBI, EVI and SAVI, and the names of the three types of LULC can be written in the title position. The abscissa range is not uniform, which should make it more intuitive to see who is better correlated.

Reply: According to reviewer’s suggestions, the modification has been done for Figure 9-12. Kindly refer to “Figure 9-12” of the “Revised Manuscript”. The abscissa ranges for types of LULC cannot be kept the same as the range of values of types of LULC varies with each index. For example, in the case of NDBI, the built-up values are from 0 to 0.8, whereas for NDVI, the built-up values are from -0.8 to 0. The figure analysis will be difficult if the abscissa range is maintained uniformly between -1 and 1.

  • I think “Figure 8” in line 401 should be “Figure 9”.

Reply: Authors agree with the reviewer’s comment, and modification has been done in the Revised Manuscript. “Figure 12 shows a weak and positive spatial association between aerosol quantity and soil-adjusted vegetation index in the scatterplots between AOD and SAVI.” Kindly refer to the “Revised Manuscript.”

Reviewer 3 Report

The authors investigated the spatial variation and relationship of aerosol optical depth (AOD) with spectral indices derived from Land Use Land Cover (LULC) from 2010 to 2019. The correlation between AOD and spectral index is used to investigate this relationship. The study's focus region (parts of Delhi and the National Capital Region) is highly urbanized areas dominated by heavy industrial pollution, vehicle emissions, fossil fuel burning, and anthropogenic activities. Using aerosol products from the Moderate Resolution Imaging Spectroradiometer (MODIS), this study sheds new light on how aerosol variation responds to LULC patterns at the regional scale.

The authors discovered a positive correlation between the Normalized Difference Built-up Index (NDBI) and the Soil Adjusted Vegetation Index (SAVI). Based on the above finding, the authors concluded that regional development has increased aerosols and air pollution levels on a mesoscale. The thorough and well-written manuscript reflects the project's comprehensive work. The manuscript is well-written and relevant to the field. The methodology, findings, and discussion are all well-represented. The manuscript's conclusions appear statistically significant and reproducible given the necessary resources. The conclusions are consistent with the evidence presented in the paper's analysis section.

The English language and the grammar presented in the paper are consistent and good. Some typing errors remain, but the changes required are all minor edits.

I have some questions and general and specific comments that I would appreciate if the authors' answered/addressed.

Line 21-23: Is the aerosol concentration in the study area high throughout the year or specific months of the year?

Line 33: Just to maintain the standard uniformity, PM2.5 (2.5 can be subscript) throughout the text.

There is no mention of MODIS in the abstract. Maybe it can be mentioned there.

Line 40: anthropogenic instead of human activities

Line 43: monsoon instead of monsoonal

Line 50-53: LULC is a very broad terminology; more elaboration is required.

Line 62-63: The sentence needs to be rephrased

Line 66: Moderate Resolution Imaging Spectroradiometer (MODIS)

Line 64-76: This paragraph needs to be further elaborated. What related research is already done, and what research gaps need to be filled in?

Equation (1-6): Are the red brackets intentional, or are they typing errors?

Table 1: It will be ideal to have the entire table on the same page

Line 252-253:  Elaboration required.

Figure 2, 7-9: Hard to read from the figure. Kindly increase the overall dimension of the figure and the font size.

Line 362-65: Concerning table 5, some more explanation is required.

Given the number of acronyms used in the manuscript, I strongly advise the authors to include a list of abbreviations at the end of the paper (before references).

Author Response

Reviewer 3

The authors investigated the spatial variation and relationship of aerosol optical depth (AOD) with spectral indices derived from Land Use Land Cover (LULC) from 2010 to 2019. The correlation between AOD and spectral index is used to investigate this relationship. The study's focus region (parts of Delhi and the National Capital Region) is highly urbanized areas dominated by heavy industrial pollution, vehicle emissions, fossil fuel burning, and anthropogenic activities. Using aerosol products from the Moderate Resolution Imaging Spectroradiometer (MODIS), this study sheds new light on how aerosol variation responds to LULC patterns at the regional scale.

The authors discovered a positive correlation between the Normalized Difference Built-up Index (NDBI) and the Soil Adjusted Vegetation Index (SAVI). Based on the above finding, the authors concluded that regional development has increased aerosols and air pollution levels on a mesoscale. The thorough and well-written manuscript reflects the project's comprehensive work. The manuscript is well-written and relevant to the field. The methodology, findings, and discussion are all well-represented. The manuscript's conclusions appear statistically significant and reproducible given the necessary resources. The conclusions are consistent with the evidence presented in the paper's analysis section.

Reply: We appreciate your precious time reviewing our paper and providing valuable comments. Your valuable and insightful comments led to possible improvements in the current version. Thank you encouraging and suggestions. We revised the whole paper per your suggestion; mistakes and confusing statements have been improved in the revised manuscript. Thank you again for your valuable suggestion and comments.

The English language and the grammar presented in the paper are consistent and good. Some typing errors remain, but the changes required are all minor edits.

Reply: We have edited the entire manuscript and improved the language, typing errors, etc, accordingly.

I have some questions and general and specific comments that I would appreciate if the authors' answered/addressed:

  • Line 21-23: Is the aerosol concentration in the study area high throughout the year or specific months of the year?

Reply: The aerosol concentration is typically high in some months of the year.

  • Line 33: Just to maintain the standard uniformity, PM2.5 (2.5 can be subscript) throughout the text.

Reply: According to the reviewer's suggestions, PM2.5 has been written as PM2.5 in the whole manuscript. Kindly refer to the "Revised Manuscript".

  • There is no mention of MODIS in the abstract. Maybe it can be mentioned there.

Reply: According to reviewer's suggestions the Moderate Resolution Imaging Spectroradiometer (MODIS) has been mentioned in the abstract. Kindly refer to the "Revised Manuscript".

  • Line 40: anthropogenic instead of human activities.

Reply: According to the reviewer's suggestions, human activities have been changed to anthropogenic ones. "Natural processes or anthropogenic activities can lead to the formation of these pollutants." Kindly refer to "line 40" of the "Revised Manuscript".

  • Line 43: monsoon instead of monsoonal.

Reply: According to reviewer's suggestions, the monsoonal pattern has been changed to monsoon patterns. Kindly refer to "line number 43" of the "Revised Manuscript".

  • Line 50-53: LULC is a very broad terminology; more elaboration is required.

Reply: According to reviewer's suggestions, the elaboration has been done in the revised manuscript. "Urbanization and economic growth leading to land use land cover (LULC) trans-formation mainly increase in the built-up area followed by fallow/open land and decrease in vegetation cover, agricultural land, and water bodies have resulted in increased emissions of air pollutants, resulting in a worsening of air quality, affecting regional climate and thereby influencing air pollution transport and diffusion [13–16]. Mainly the increase in infrastructure leads to built-up (high-rise buildings, roads, highways etc.) increment and decrement of cropland [17]." Kindly refer to the "paragraph no 2 of Section 1: Introduction" of the "Revised Manuscript".

  • Line 62-63: The sentence needs to be rephrased.

Reply: As per the reviewer's suggestions, the changes have been changed: "By using remote sensing, we can solve the gaps created by the absence or dispersion of weather observatories." Kindly refer to "Section 1" and "last line of paragraph no 3" of the "Revised Manuscript".

  • Line 66: Moderate Resolution Imaging Spectroradiometer (MODIS).

Reply: According to reviewer's suggestions, the changes have been incorporated. Kindly refer to the "line number 66" of the "Revised Manuscript".

  • Line 64-76: This paragraph needs to be further elaborated. What related research is already done, and what research gaps need to be filled in?

Reply: According to the reviewer's suggestions, the Revised Manuscript has elaborated the paragraph. Kindly refer to the "paragraph no 5 of Section 1: Introduction" of the "Revised Manuscript".

  • Equation (1-6): Are the red brackets intentional, or are they typing errors?

Reply: These were not the typing errors nor the red brackets intentional. Its automatically taken by the system (MS office 2022).

  • Table 1: It will be ideal to have the entire table on the same page.

Reply: According to reviewer's suggestions, the table has been modified in the "Revised Manuscript". Kindly refer to the "Table 1" of the "Revised Manuscript".

  • Line 252-253:  Elaboration required.

Reply: According to reviewer's suggestions, the elaboration has been done in the Revised Manuscript. "Approximately 93% of AOD values at Amity University and 79% at Gual Pahari fall inside the EE. This demonstrates that AERONET AOD observations and MAIAC AOD values within the 0.5 to 1.0 are more closely aligned. In the present AOD range of 0.5 to 1.0, the underestimation also decreased." Kindly refer to "paragraph no 1 of Section 4: Results" of the "Revised Manuscript".

  • Figure 2, 7-9: Hard to read from the figure. Kindly increase the overall dimension of the figure and the font size.

Reply: According to reviewer's suggestions, the modification has been done for Figure 2, 9-12. Kindly refer to the "Figure 2, 9-12" of the "Revised Manuscript".

  • Line 362-65: Concerning table 5, some more explanation is required.

Reply:  According to reviewer's suggestions, the elaboration has been done in the Revised Manuscript. "The NDVI and EVI were negatively correlated with AOD with the value of - 0.24, - 0.15, which shows that the vegetation has a purification impact on AOD. The build-up increase has also raised AOD values, according to the positive NDBI correlation (0.35). The NDVI, NDBI, EVI, SAVI, and AOD correlation coefficients were not high enough, indicating that there may not be a clear or continuous association on the city scale and that indexes alone are insufficient to explain the variance in AOD across the research region fully. Consequently, the suitable landscape context should be used for analysis to support further research."

Kindly refer to the "Section 4.2" of the "Revised Manuscript".

  • Given the number of acronyms used in the manuscript, I strongly advise the authors to include a list of abbreviations at the end of the paper (before references).

Reply: According to reviewer's suggestions, the list of abbreviations has been added before references. Kindly refer to the "Revised Manuscript " list of abbreviations".

Reviewer 4 Report

The authors investigated the correlation between AOD and spectral indices including NDVI, SAVI, EVI, and NDBI, and attempted to illustrate the impacts of land use land cover on AOD and air quality. This is an interesting topic for those who are devoted to solve the problems of air pollution. This paper is well organized of most parts, but still needs several modifications to satisfy the criteria for publication.

1)     The correlation coefficients (e.g., 0.35 and 0.27) are not large enough to draw a conclusion like “built-up soils play an important role in accumulating AOD”. The authors should try to collect more data to get a stronger correlation.

2)     The last paragraph of introduction, AOD AERONET à AERONET AOD

3)     Table 1, why not use AERONET AOD of Version 3?

4)     Figure 2, the subfigures are too small to get clear spatial distribution of AOD

5)  Figure 7, 8, and 9, the title of x-axis of subfigures should be NDBI, EVI and SAVI.

Author Response

Reviewer 4

The authors investigated the correlation between AOD and spectral indices including NDVI, SAVI, EVI, and NDBI, and attempted to illustrate the impacts of land use land cover on AOD and air quality. This is an interesting topic for those who are devoted to solve the problems of air pollution. This paper is well organized of most parts, but still needs several modifications to satisfy the criteria for publication.

Reply:

  • The correlation coefficients (e.g., 0.35 and 0.27) are not large enough to draw a conclusion like “built-up soils play an important role in accumulating AOD”. The authors should try to collect more data to get a stronger correlation.

Reply: The authors agree with the reviewer's assessment that these correlation coefficients are too low to draw any firm conclusions. As a pilot study for thesis work, the present study will be part of a larger-scale research project. Additionally, the authors have determined that landscape context research is necessary for this, as done by Xie et al. in 2021 [6] since indices alone are insufficient to draw attention to the city scale.

  • The last paragraph of introduction, AOD AERONET à AERONET AOD.

Reply: According to reviewer’s suggestions, the term has been changed to ARONET AOD. Kindly refer to “line 110” of the “Revised Manuscript”.

  • Table 1, why not use AERONET AOD of Version 3?

Reply: Version 2 was written due to typographical error and the modification has been done in the “Revised Manuscript”. Kindly refer to the “Table 1” of the “Revised Manuscript”.

  • Figure 2, the subfigures are too small to get clear spatial distribution of AOD.

Reply: According to reviewer’s suggestions Figure, 2 has been changed. Kindly refer to the “Figure 2” of the “Revised Manuscript”.

  • Figure 7, 8, and 9, the title of x-axis of subfigures should be NDBI, EVI and SAVI.

Reply: According to the reviewer’s suggestion, the x-axis has been modified. Kindly refer to “Figure 9-12” of the “Revised Manuscript”.

Round 2

Reviewer 2 Report

I am happy to accept the paper as the authors have addressed all my comments. Thanks for the efforts.

Reviewer 4 Report

The authors did make substantial corrections so the paper has clearly improved in the revision. It seems to me that the figures and text of the paper are better after the revisions. The authors have also answered to all the comments or questions in the response. This paper can be accepted for publication.